# Enhancing Communication Efficiency and Training Time Uniformity in Federated Learning through Multi-Branch Networks and the Oort Algorithm

Pin-Hung Juan [1] and Ja-Ling Wu [1,2,3,*]

1 Department of Computer Science and Information Engineering, National Taiwan University,
Taipei 106, Taiwan
2 Graduate Institute of Networking and Multimedia, National Taiwan University, Taipei 106, Taiwan
3 Center for Data Intelligence: Technologies, Applications, and Systems, National Taiwan University,
Taipei 106, Taiwan
* Correspondence: wjl@cmlab.csie.ntu.edu.tw

**Abstract:** In this study, we present a federated learning approach that combines a multi-branch network and the Oort client selection algorithm to improve the performance of federated learning systems. This method successfully addresses the significant issue of non-iid data, a challenge not adequately tackled by the commonly used MFedAvg method. Additionally, one of the key innovations of this research is the introduction of uniformity, a metric that quantifies the disparity in training time amongst participants in a federated learning setup. This novel concept not only aids in identifying stragglers but also provides valuable insights into assessing the fairness and efficiency of the system. The experimental results underscore the merits of the integrated multi-branch network with the Oort client selection algorithm and highlight the crucial role of uniformity in designing and evaluating federated learning systems.

**Keywords:** federated learning; uniformity; communication efficiency; client selection; multi-branch network

## 1. Introduction

Federated Learning (FL) [1] has emerged as a powerful approach for training machine learning models on decentralized data without compromising data privacy. It allows multiple clients to collaboratively train a shared global model while keeping their data locally. This distributed learning paradigm has gained significant attention and has been applied to various domains, including healthcare, finance, and the Internet of Things (IoT).

The primary objective of federated learning is to improve communication efficiency and ensure uniform training times among clients; however, the heterogeneity of data and systems in federated learning challenges client selection and training processes. Selecting appropriate clients to participate in training becomes crucial to achieving accurate and efficient model updates.

In this context, the Oort algorithm has been proposed as a client selection method that considers the heterogeneity of data and systems. However, implementing the Oort algorithm (detailed in Section 3.3) revealed temporal discrepancies in training and communication, leading to inefficient federated learning.

We propose integrating the Multi-Branch Network (MBN) into the existing Oort architecture to address this issue and enhance communication efficiency and training uniformity. The MBN construction is inspired by BranchyNet [2] and Triplewins [3], where additional branch classifiers are incorporated at equidistant points within a given neural network. This modification allows for model averaging and improved performance without needing multiple convolutional layers in each branch.

Furthermore, we introduce a Model Distributor component to allocate different model branches to clients based on their computational capabilities and communication bandwidth. This clustering-based approach ensures that clients receive models tailored to their specific training requirements, optimizing the effectiveness and efficiency of the federated learning process.

This work presents the construction of the MBN and the Model Distributor in detail. We evaluate the performance of our proposed method on benchmark datasets, including CIFAR-10 and FEMNIST, using the ResNet34 model and the MBN. We measure the training time to achieve specific accuracy levels and the number of rounds required to reach the desired performance. Additionally, we analyze the uniformity of the training process and investigate the impact of different hyperparameters.

The rest of this paper is organized as follows: Section 2 provides background information on federated learning and MBN. Section 3 explores the related work in federated learning. Section 4 describes the construction of the MBN and the Model Distributor. Section 5 presents the experimental results and discusses the findings. Finally, Section 6 concludes the thesis and outlines potential directions for future research.

## 2. Preliminary

### 2.1. Federated Learning

Federated Learning has emerged as a promising approach in machine learning, enabling decentralized training while addressing privacy and data ownership concerns. The Federated Averaging (FedAvg) algorithm proposed by McMahan et al. [1] is widely used for global aggregation in FL.

FL leverages the power of local devices such as smartphones and tablets to perform model training while a central server aggregates the locally computed updates [4]. This distribution of the learning process brings several advantages [5,6]. Firstly, it mitigates privacy risks by avoiding transferring sensitive data to a central location. Secondly, FL allows the utilization of device-specific data that would otherwise be challenging to access due to privacy or logistical constraints [7–9].

However, FL also presents its inherent challenges [6,10]. One such challenge is data heterogeneity, where the data distribution across different devices may vary significantly. This heterogeneity can affect the convergence and performance of FL models.

Another challenge is the heterogeneity of computing resources among client devices. Some clients may have limited computational capabilities or unreliable network connections, which can lead to stragglers, slowing down the overall FL process.

Moreover, communication efficiency is another crucial factor in FL. Since clients must frequently communicate with the central server to obtain the latest model updates, efficient communication protocols and strategies are necessary to reduce communication overhead.

In the upcoming section, we will delve into various research works that aim to tackle these challenges, including data and computing resource heterogeneity and communication efficiency.

### 2.2. Multi-Branch Networks

The concept of multi-branch networks was first introduced in [2] and further developed in [3]. In contrast to traditional neural networks, which only have a single exit point, multi-branch networks are designed to incorporate multiple exit points. This architecture features numerous early-branch output layers and the standard final output layer, enhancing the network's capability to capture and leverage diverse intermediate representations from various branches. As a result, it enhances performance and versatility in handling intricate tasks.

The early-branch output layers within a multi-branch network facilitate the extraction of specific features or representations at intermediary stages of network processing. By offering auxiliary outputs or intermediate predictions, these layers contribute significantly

to guiding the network's learning process and provide additional regularization to fortify the network's stability and generalization.

Considering the benefits of multi-branch networks, it is advantageous to employ them as the training model in Federated Learning environments [11]. Their architectural features lend to enhanced learning and adaptability, making them a suitable choice for the diverse and distributed nature of FL systems.

## 3. Related Work

Several academic papers have put forth various methodologies to address the challenges mentioned in the preliminary section regarding Federated Learning. These approaches can be broadly categorized into homogeneous model FL and heterogeneous model FL. Table 1 summarizes the characteristics and limitations of each considered related work. Moreover, a comparison of the reduction in non-iid impacts, alleviation in communication bandwidth, alleviation in computational capability, and improvement in transfer speed among the benchmarking works is presented in Table 2.

**Table 1.** Summarizations of the characteristics and limitations of the benchmarked federated learning techniques.

| Name of the Method | Characteristics | Limitations |
|---|---|---|
| FedProx [12] | Add a proximal term to ensure that the local models of participants stay close to the global model. | It is challenging to select an appropriate center point $\omega_c$ for the proximal term. |
| FedYogi [13] | Add a gradient correction to further suppress the data heterogeneity and performance variations among participants. | The computation cost increases with the calculation of gradient corrections, $\Delta_{\omega_k}$. |
| FedTCR [14] | 1. Groups clients by resources. <br> 2. Only the group leader communicates with the server. | 3. Only considers the computing resources. <br> 4. Privacy issues arise. |
| FedTiny [15] | Select pruned models by evaluating client datasets and further sparsify the update parameters. | The pruning procedure incurs additional computation and transmission costs. |
| FedDF [16] | Utilize knowledge distillation to share information across different types of models. | It needs a public dataset that is unrealistic for real-world scenarios. |
| MFedAvg [11] | Distribute models of different sizes to clients, allowing each client to receive and accommodate a suitable model. | It does not consider the scenario of non-iid data. |

**Table 2.** Comparisons of benchmarked federated learning techniques (in which the symbol "O" denotes the issue that has been addressed in the method).

| Method//Issues | Reduce Non-iid Impact | Alleviate Communication Bandwidth | Alleviate Computational Capability | Improve Transfer Speed |
|---|:---:|:---:|:---:|:---:|
| FedProx [12] | O | | | |
| FedYogi [13] | O | | O | |
| FedTCR [14] | | | O | O |
| FedTiny [15] | O | | | O |
| FedDF [16] | O | O | O | |
| MFedAvg [11] | | O | O | O |
| Oort [17] | O | O | O | |

### 3.1. Homogeneous Model FL

To address the challenge of non-iid (non-independent and identically distributed) data in FL, Li et al. proposed FedProx [12]. It introduces a proximal term to the FedAvg

algorithm, a commonly used algorithm in FL for aggregating local model updates from participant devices. The proximal term in FedProx aims to keep the local models of participants close to the global model by imposing a penalty if the local data are biased. This penalty encourages participants to contribute updates that align with the global model. Building upon FedProx, Reddi et al. proposed FedYogi [13]. FedYogi enhances FedProx by introducing gradient corrections. These corrections consider the performance variations among participants and adaptively adjust the importance of each participant's parameter updates to the server. By considering the individual participants' performance, FedYogi aims to better use the updates from participants with higher reliability or accurate data while reducing the impact of updates from participants with less reliable data. However, calculating the gradient corrections in FedYogi can introduce additional computation overhead, potentially increasing the training time.

By minimizing the variability in total computing resources within each group, Fed-TCR [14] aims to tackle the challenge of resource heterogeneity among participants in federated learning. This approach helps ensure that every group can collectively contribute to the training process without significant discrepancies in computing capabilities. Only the client with the most substantial computing resources in each group can directly communicate with the server. At the same time, the remaining participants exchange the model update with the cluster head to alleviate the communication overhead on the server. While this architecture can reduce communication costs and address the heterogeneity of computing resources, it may introduce privacy concerns as the trustworthiness of the cluster head is not guaranteed.

As an intuitive approach to reducing communication costs, FedTiny [15] introduces a unique method to address the challenge of non-iid data. It achieves this by employing model pruning techniques. FedTiny creates multiple pruned models and allows participants to update the batch normalization layer to analyze their data distribution indirectly. The server can select a model with a minimum bias for each participant from the candidate model pool. However, this approach still suffers from a decrease in overall accuracy due to discarding specific parameters during the pruning process.

### 3.2. Heterogeneous Model FL

In contrast to federated learning with homogeneous models, FedDF [16] employs knowledge distillation to extract logits from participants [18]. By obtaining the logits, the server can update prototype models on the server side, eliminating the need for participants to update the parameters of their local models. This approach allows FedDF to accommodate the heterogeneity of model settings, enabling variations in model architectures among participants. However, it should be noted that FedDF requires a proxy dataset to perform the distillation process, which may be unrealistic in real-world scenarios.

On the other hand, MFedAvg [11] utilizes a multi-branch network to leverage federated learning with heterogeneous models. By assigning the early exit branch to weak clients and the whole model to substantial clients, MFedAvg effectively mitigates the discrepancy in computation capability and communication bandwidth among clients. Unfortunately, MFedAvg does not address the impact of non-iid data, which is a limitation.

### 3.3. Oort—Client Selection for FL

As predescribed, the existing FL works optimize for better training accuracy with fewer training rounds (the so-called statistical model efficiency) or shorter average time duration per round (the so-called system efficiency), in which the participating clients are randomly selected for ease of deployment. Nevertheless, as pointed out by [17], a random selection of participants may lead to poor performances, biased testing sets, and loss of confidence in FL results.

Unlike previous works that address specific challenges, Oort [17] proposes a client selection framework (located inside the coordinator of an FL framework and interacting with the driver of an FL execution) to select high-quality clients for effective participation

in the training job. It utilizes a utility function to measure the clients' priority based on three dimensions: data distribution, computing resources, and communication bandwidth. The system architecture of Oort is depicted in Figure 1.

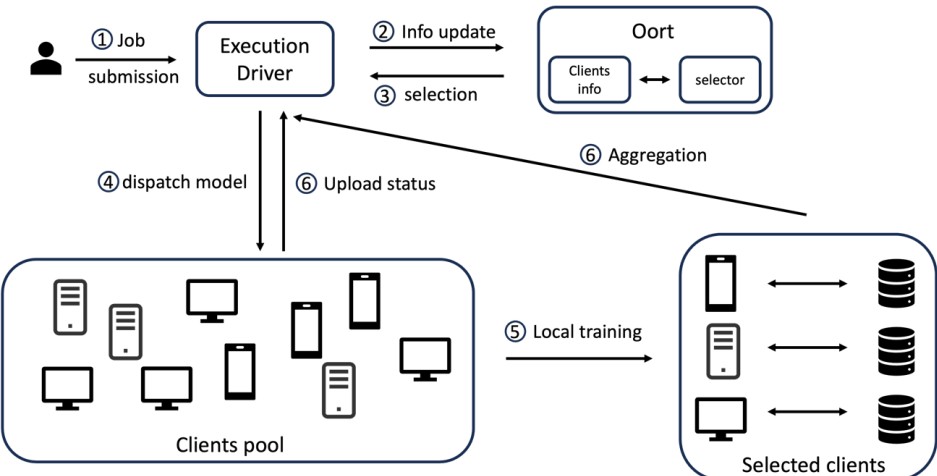

**Figure 1.** The system architecture of Oort (the indicated numbers represent the sequence order of the Oort's execution).

To make the extra cost paid for selection affordable, the authors of [17] (Section 4.2) introduced an effective statistical utility measure for each client, which can capture the heterogeneous data utility across and within categories and samples for various tasks. In other words, Oort's selection framework can also consider the non-iid client data issue by defining an appropriate utility measure. Moreover, [19] presented theoretical proof of the effectiveness of the adopted utility function in [17] over random sampling and empirically justified its performance in practice.

However, even if the Oort framework selects preferable clients, there may still be a significant training time gap among the selected clients. According to our experimental results, as the green numbers in Figure 2 show, the fastest client needed to wait for the slowest client for nearly 1400 s to complete the whole training, which significantly burdened the overall training performance. This surprising observation inspired us to investigate ways to enhance the overall efficacy of FML.

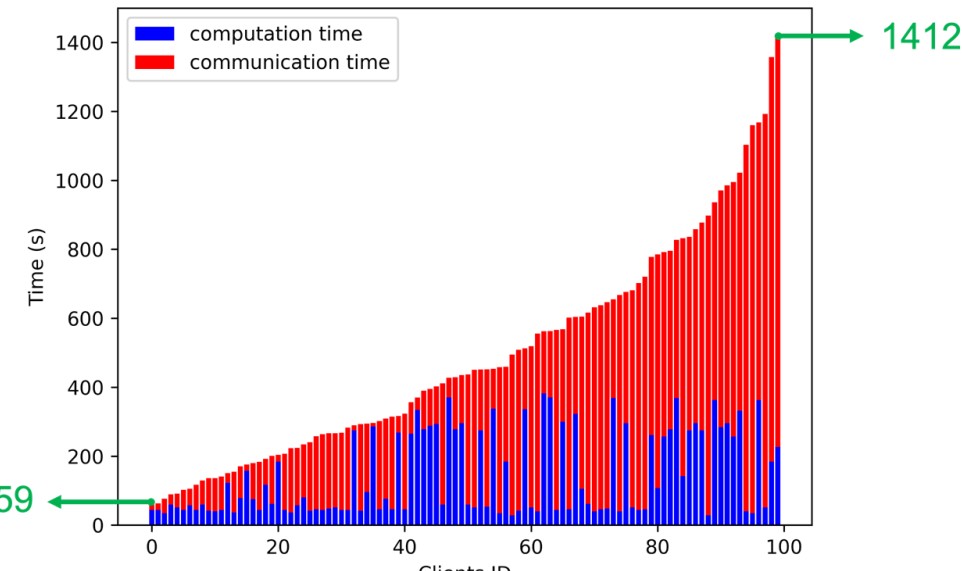

**Figure 2.** The overall training time gap among selected clients using Oort (in which the experiment conducted followed the environment settings presented in Section 5.1).

### 3.4. Federated Machine Learning in Edge Computing and Wireless Communications

As reminded by the anonymized reviewers, besides wireline connected systems (as considered in our work), FL has also been widely applied in edge computing [20–23], where users are connected to an edge server via wireless links, to achieve ubiquitous intelligence. However, the limited uplink capacity of wireless channels handicaps the convergence in edge FL's model aggregation, making a long convergence time unavoidable. Many effective edge FL designs [24–29] have been proposed to conquer the abovementioned challenge. For example, [28] offered a Unit-Modulus Over-the-Air Computation (UMAirComp) framework to facilitate efficient edge FL, which simultaneously uploads local model parameters and updates global model parameters via analog beamforming. Simulation results justified that UMAirComp achieved a minor mean square error of model parameters' estimation, training loss, and test error compared with other benchmark schemes.

Moreover, as addressed in [29], multiple distinct datasets are usually generated from massive IoT users and correspond to different learning tasks. Therefore, designing parallel and low-complexity algorithms for various learning tasks becomes imperative for large-scale IoT networks. Specifically, concurrent transmissions among massive terminals in large-scale IoT networks will inevitably yield severe Co-Channel Interference (CCI), greatly degrading system performance. To deal with the CCI, [29] designed a multi-user scheduling algorithm to mitigate the CCI issue. Moreover, [29] also developed a parallel algorithm to solve the power allocation for different tasks. We want to keep the focus of our discussion, so we refer readers interested in edge FL and task-oriented FL to [28,29] and the references therein, respectively.

## 4. The Proposed Method

The primary objective of our proposed method is to improve communication efficiency and ensure as uniform training times as possible during the client selection process described in [17]. The Oort algorithm accounts for the heterogeneity of data and systems to identify appropriate participants. However, implementing the Oort algorithm revealed significant temporal discrepancies in training and communication. We propose integrating the multi-branch network into the existing Oort architecture to rectify this issue and promote enhanced communication efficiency along with uniform training times. As mentioned in Section 3, the MFedAvg method [11] does not explicitly address the impact of non-iid data, which is considered a limitation of the approach. On the other hand, combining the Oort algorithm and the multi-branch network, as proposed in this work, can effectively alleviate the impact of non-iid data [11].

In the rest of this section, we will first explain the construction of a multi-branch network derived from the original neural network. We will then introduce the model distributor and receiver we added to the original Oort system.

### 4.1. Construction of a Multi-Branch Network

Based on the findings of BranchyNet [2] and Triple wins [3], a multi-branch network can be constructed by incorporating additional branch classifiers at equidistant points within a given network, thereby facilitating model averaging. Furthermore, it has been observed that satisfactory performance can be achieved without the addition of multiple convolutional layers to each branch. For example, we can consider the ResNet34 architecture, a classical neural network (depicted in Figure 3). Our approach incorporates several additional convolutional layers into the residual blocks at every two blocks (as illustrated in Figure 4). This modification enhances the architecture by introducing branch classifiers at equidistant points, allowing for model averaging and improved performance.

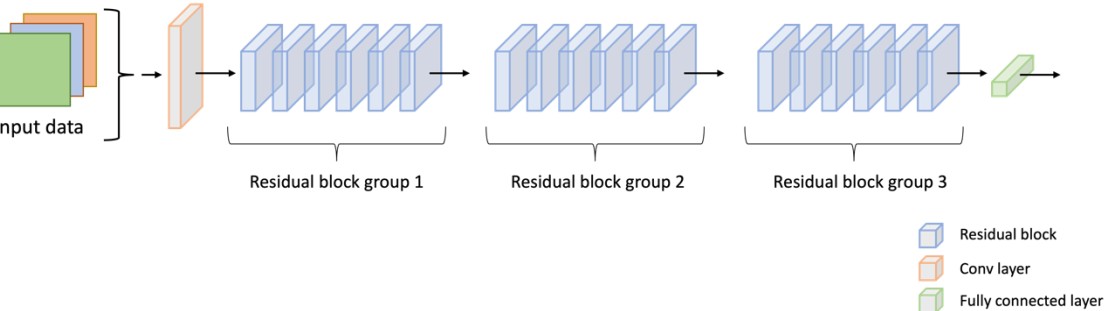

**Figure 3.** The architecture of the original ResNet34 network.

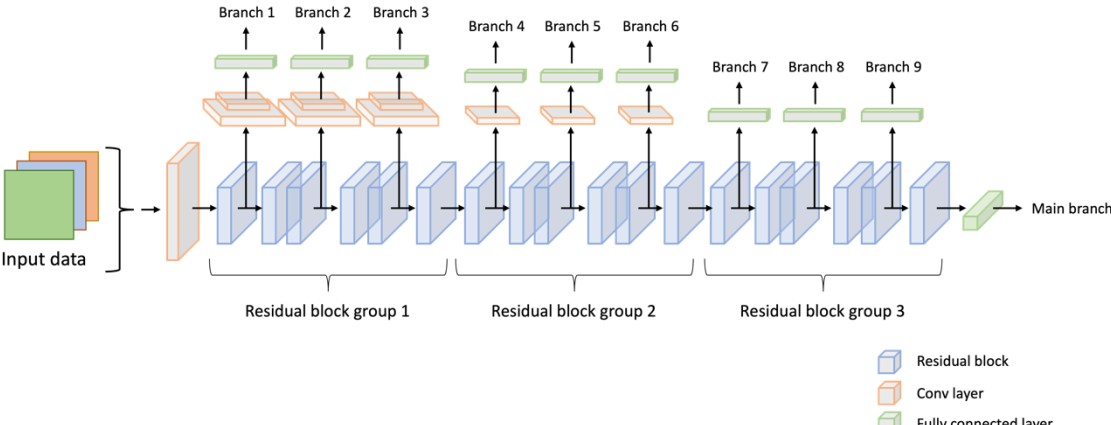

**Figure 4.** The architecture of the proposed nine-branch ResNet34 network.

### 4.2. Model Distributor

Our method presents an alternative to the conventional federated learning paradigm, wherein clients are served identical models from the server. Our strategy allocates distinct model branches to various clients based on their computational capabilities and communication bandwidth. To facilitate this, we introduce a clustering algorithm that groups the clients into $K + 1$ clusters, where $K$ denotes the number of additional branches incorporated into the model.

The initial step in the tier clustering algorithm (Algorithm 1) involves assigning the overall training capability by considering each client's computation capability and communication bandwidth. It is worth noting that a coefficient μ is introduced to the computation capability (Line 3), where μ represents the ratio of computation capability to communication bandwidth. This coefficient determines the relative importance of computation capability concerning communication bandwidth, with μ > 1 indicating a higher emphasis on computation capability and μ < 1 indicating a higher emphasis on communication bandwidth. When μ is equal to 1, both factors are considered equally important. Section 5 will explore the diverse outcomes achieved by employing different values of μ. Furthermore, the clients are sorted (Line 5 in Algorithm 1) based on their training capability after assigning the overall training ability. They are subsequently grouped (Line 8 in Algorithm 1) into K + 1 groups, arranged in ascending order according to their training times, starting from the clients with the lower training capability and progressing toward those with more substantial training capabilities.

With the clustering set in place, the model distributor can assign different models based on the individual training capabilities of clients. The branch models $\{M_i\}_{i=1}^{K+1}$ represent the initial neural network architecture's early exit points. Each $M_i$ signifies a specific model configuration, where $i = 1$ corresponds to the most miniature model denoting the earliest exit of the network. Conversely, $i = k + 1$ represents the complete network without any early exits.

The model distributor systematically dispatches the corresponding model $M_i$ to the clients within cluster $G_i$ to optimize the training process. This sequential assignment ensures that clients receive a model tailored to their specific training requirements, ultimately maximizing the effectiveness and efficiency of the federated learning process.

---

**Algorithm 1: Tier Clustering Algorithm**

---

**Require:** Clients set $\varkappa = \{\varkappa i\}_{i=1}^{N}$, computation capability set $\mathbf{P} = \{Pi\}_{i=1}^{N}$, communication bandwidth set $\mathbf{S} = \{Si\}_{i=1}^{N}$, clustering set $G = \varnothing$, ratio of computation capability to communication bandwidth $\mu$, number of additional branches **K**
**Ensure:** Clustering set $G$
1:  Training capability set $T \leftarrow \varnothing$
2:  **for** $i = 1$ to N **do**
3:      Update Training capability set $T_i = \mu * P_i + S_i$
4:  **end for**

5:  Sort the set $T$ and obtain the sorted set $T' = \{T_1' < T_2' < ... , < T_N'\}$ and corresponding client set $\varkappa' = \{\varkappa i\}_{i=1}^{N}$

6:      **for** t = 1 to K + 1
7:          **for** j = t to t + N/(K + 1) **do**
8:              Sequentially assign $\varkappa'_j$ to $G_t$
9:          **end for**
10:     **end for**

---

### 4.3. Overall System Architecture

We implemented our work using FedScale [30], an open-source evaluation platform and benchmark designed explicitly for federated learning. Figure 5 illustrates the overall system architecture of our implementation.

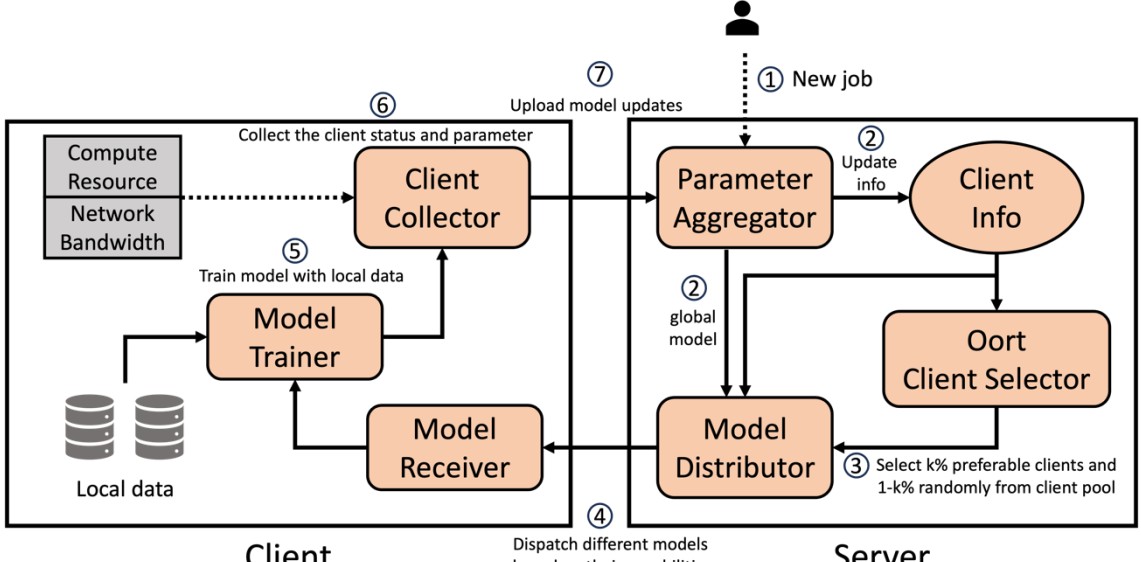

**Figure 5.** The overall system architecture and the functional block diagram of our proposed framework.

In this architecture, the user first submits the job, which includes the hyperparameter settings (e.g., clients per round and μ), to the Parameter Aggregator (acting as the primary server) ①.

Additionally, the Parameter Aggregator receives feedback from the clients regarding the previous training round, updates the global model with the clients' updates, and collects relevant information from the clients ②.

Afterward, the Oort Client Selector will select a certain percentage (k%) of preferable clients based on their computing capability, communication bandwidth, and data distribution. The remaining quota will be randomly selected from the pool of unselected clients ③.

The Model Distributor receives the aforementioned information, including the updated global model, client information, and the selected clients. It utilizes the Tier Clustering Algorithm to group the selected clients into several clusters based on their characteristics. Then, it dispatches the suitable model to each client within their respective cluster, ensuring an efficient and tailored training process ④.

The clients receive the assigned models through the Model Receiver and train with their local data in the Model Trainer ⑤.

After completing the training process, the Client Collector collects the local model parameters and captures their corresponding status ⑥.

Finally, the collected data, including the model parameters and their status, are transmitted and updated on the server ⑦.

The entire process is executed iteratively, with periodic testing every few rounds until the desired number of target rounds is reached.

## 5. Experiments

In this section, we discuss the experiments we performed on well-known benchmark datasets, including CIFAR-10, CIFAR-100 [31], and FEMNIST [32], with the ResNet-34 model [33] and the Multi-Branch ResNet-34 (MB_ResNet-34). We present the experimental results in terms of two metrics: the time taken to achieve a specific accuracy (time to accuracy) and the number of rounds required to reach a certain level of accuracy (rounds to accuracy).

### 5.1. Experimental Setup

#### 5.1.1. Environment Settings

Our experiments were conducted using the FedScale [30] platform, and we ensured a consistent environment setting across different dataset experiments. The client pool consisted of 2800 clients, with 100 clients selected in each round of training. We performed a total of 1000 training rounds. The datasets were divided into non-iid partitions. The computation capability of clients was predetermined prior to the training process, while the communication bandwidth of clients varied during the training process. Furthermore, the client's online/offline status also fluctuated throughout training.

#### 5.1.2. Model Settings

To construct the multi-branch ResNet-34 architecture, we followed the methodology described in [3,4]. In our implementation, we incorporated additional branches into the original ResNet-34 architecture. Specifically, we added three extra branches for the first group of residual blocks. Each branch consisted of two convolutional layers followed by a fully connected layer. These additional branches were inserted at regular intervals of every two consecutive blocks within the first group.

Similarly, we added branches combining one convolutional layer and a fully connected layer for the second group of residual blocks, following the same pattern of regular intervals. Finally, we added a single fully connected layer without any convolutional layers for the last group of residual blocks. The complete network structure, including the additional branches, is illustrated in Figure 4.

### 5.2. Training Details

We want to clarify that Section 5.1.1 does not explicitly mention the specific hyperparameter settings such as local training steps, batch size, and learning rate. However, it should be noted that these hyperparameters were consistent with the settings specified in the Oort [17].

We followed the training process outlined in [17] for training the ResNet-34 model. However, there was a crucial difference in the training of the multi-branch ResNet-34 model. After selecting clients using the Oort selection algorithm, we clustered them into ten groups, corresponding to the nine additional branches and the main branch. We then assigned the respective branch models to each client group for training. The detailed training process, including the Tier Clustering Algorithm, can be found in Section 4.3.

*5.3. Experimental Results*

To evaluate the effectiveness of our approach, we conducted experiments on the FEMNIST and CIFAR-10 datasets using different model settings. We assessed the performance using two primary metrics: time-to-accuracy and rounds-to-accuracy. These metrics provide insights into the efficiency and effectiveness of our approach in achieving accurate results within a given time frame and number of training rounds.

5.3.1. Time-to-Accuracy Performance

Our experiments evaluated the time taken to achieve accuracy levels of 60%, 70%, and 80%. Table 3 presents the experimental results concerning the FEMNIST dataset. The first row represents the performance of the ResNet34 model without the Oort selection algorithm. It took approximately 66,761 s to achieve 60% accuracy and 101,072 s to achieve 70% accuracy. In contrast, the MB_ResNet34 model achieved the same level of accuracy in significantly less time than the original ResNet34 model.

**Table 3.** Training time versus accuracy for different models on the FEMNIST dataset (in which the symbol "−" means that accuracy was not achievable due to the impact of non-iid data during the experiment).

| Model | Accuracy 60% | Accuracy 70% | Accuracy 80% |
|---|---|---|---|
| ResNet34 | 66,761 | 101,072 | — |
| MB_ResNet34 | 6,799 | 13,097 | — |
| ResNet34 + Oort | 47,701 | 68,869 | 286.324 |
| MB_ResNet34 +Oort | 5,555 | 10,113 | 36.854 |

However, it should be noted that the scenarios without the client selection algorithm faced poor overall accuracy due to the impact of non-iid data. To address this issue, we integrated the selection algorithm, as shown in the third and fourth rows of the table. It is evident that, compared to the random selection method, the time to accuracy decreased, and the overall accuracy exceeded 80%.

For the CIFAR-10 dataset, as shown in Table 4, we observed similar trends to those in Table 3. This observation further supports the effectiveness of our proposed method in alleviating the negative impact of non-iid data on the MB_ResNet34 model in [11] (rows 2 and 4) and reducing the training time compared to the original ResNet34 model in [17] (rows 3 and 4).

**Table 4.** Training time versus accuracy for different models on the CIFAR-10 dataset (in which the symbol "−" means that accuracy was not achievable due to the impact of non-iid data during the experiment).

| Model | Accuracy 60% | Accuracy 70% | Accuracy 80% |
|---|---|---|---|
| ResNet34 | 246,102 | 837,236 | — |
| MB_ResNet34 | 23,036 | 48,558 | — |
| ResNet34 + Oort | 225,743 | 811,432 | 2,432,746 |
| MB_ResNet34 +Oort | 20,365 | 43,812 | 124,677 |

### 5.3.2. Rounds-to-Accuracy Performance

Figures 6 and 7 display the rounds to accuracy curves during 1000 rounds. These curves provide a more precise visualization of the overall accuracy trends. It is evident that the MB_ResNet34 + Oort model, represented by the red line, achieved the highest accuracy compared to the other models. On the other hand, the models trained without the selection algorithm, which correspond to blue and orange lines, , exhibited lower accuracy levels.

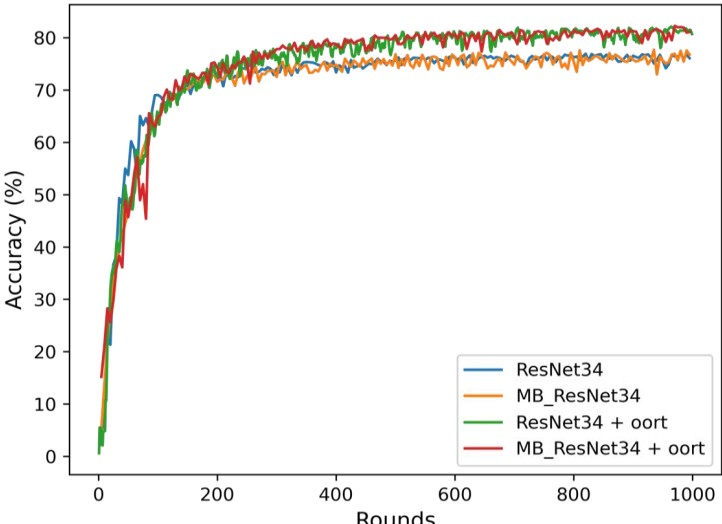

**Figure 6.** Rounds versus accuracy for different models on the FEMNIST dataset.

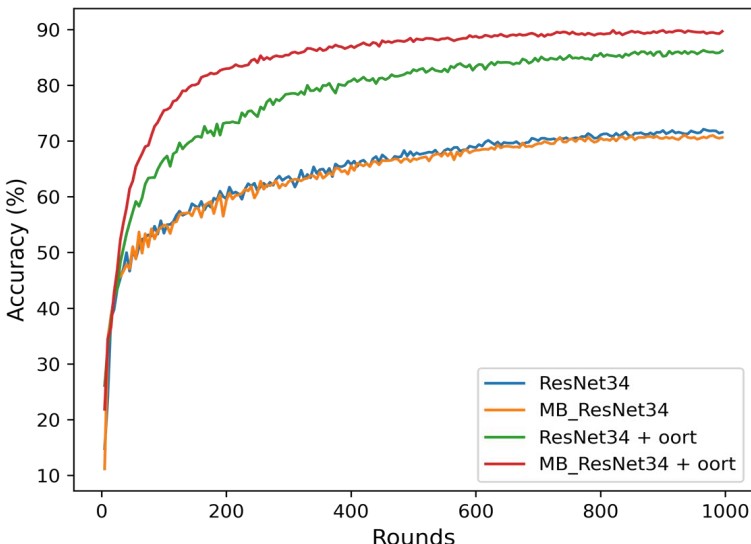

**Figure 7.** Rounds versus accuracy for different models on the CIFAR-10 dataset.

### 5.3.3. Uniformity

We also evaluated the uniformity, as defined in Equation (1), of the original ResNet34 and MB_ResNet34 models. The physical meaning of Equation (1) is similar to that of the variance but slightly modified to better capture the concept of uniformity.

$$Uniformity \triangleq \left( \frac{1}{N} \sum_{i=1}^{N} (time_i - minT)^2 \right)^{\frac{1}{2}} (1) \qquad (1)$$

In Equation (1), $time_i$ refers to the time the *i-th* client takes during the training process, and *minT* represents the minimum time spent by any client. By calculating the squared deviation of each client's time from the minimum time and then averaging them, we

measure the uniformity of the training process across selected clients. Taking the result's square root further helps provide a more interpretable value for the uniformity metric.

By utilizing the formula mentioned above, we can gain valuable insights into the consistency and uniformity of the training process for the original ResNet34 and MB_ResNet34 models, as depicted in Figures 8 and 9, respectively. As a reference, we also included the variance in total training time and the uniformity of computation time and communication time. Evidently, the uniformity significantly decreased from approximately 532 to 313 when implementing the multi-branch network. This fact demonstrates the effectiveness of the multi-branch network in improving the uniformity and consistency of the training process.

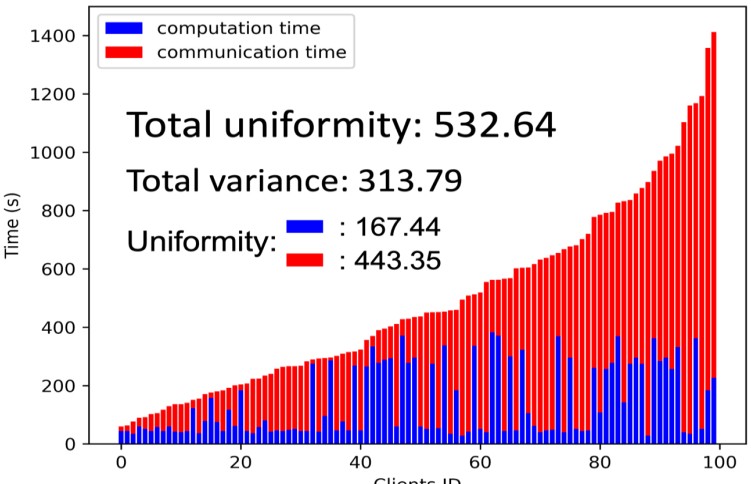

**Figure 8.** The uniformity of the original ResNet34.

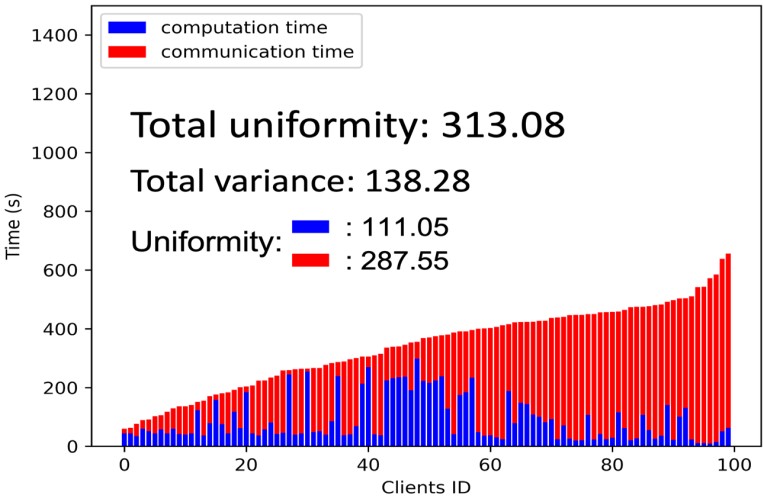

**Figure 9.** The uniformity of the MB_ResNet34.

### 5.4. Ablation Studies

5.4.1. Integration with Other Methods

In addition to conducting experiments with ResNet34 and MB_ResNet34 on CIFAR-10 and FEMNIST datasets, we also explored their performance in combination with the FedProx method [12] and FedYogi method [13]. These methods were employed to address the impact of non-iid data, reducing the time required to achieve the desired accuracy. Table 5 presents the training time-to-accuracy performance for the two models combined with different gradient policies. It is important to note that all experiments in the ablation study were performed with Oort client selection.

**Table 5.** Training time versus a certain amount of accuracy for different models on CIFAR-10 and FEMNIST datasets.

| Model Accuracy | CIFAR-10 | | | FEMNIST | | |
|---|---|---|---|---|---|---|
| | **60%** | **70%** | **80%** | **60%** | **70%** | **80%** |
| **ResNet34** | 225,743 | 811,432 | 2,432,746 | 47,701 | 68,869 | 286,324 |
| **MB_ResNet34** | 23,036 | 48,558 | 131,692 | 5555 | 10,113 | 36,854 |
| **ResNet34 + Prox** | 217,639 | 824,295 | 2,142,973 | 29,639 | 48,454 | 168,844 |
| **MB_ResNet34+ Prox** | **20,365** | **43,812** | **124,677** | **4563** | **6224** | **25,192** |
| **ResNet34 + Yogi** | 223,420 | 803,822 | 2,342,924 | 44,845 | 74,458 | 253,798 |
| **MB_ResNet34 + Yogi** | 55,222 | 98,168 | 200,200 | 4013 | 5942 | 16,530 |
| **MobileNet_v2** | 56,950,432 | — | — | | | |
| **MB_MobileNet_v2** | 33,457 | 60,464 | 138,418 | | | |

The results indicate that integrating MB_ResNet34 with the FedProx method yielded the best performance. However, when combined with the FedYogi method, MB_ResNet34 required more time than its standalone version. This additional time can be attributed to the computation overhead in calculating the gradient corrections. Furthermore, we extended our experiments to the CIFAR-100 dataset, and the results in Table 6 demonstrate a reduction in training time due to better uniformity among selected clients.

**Table 6.** Training time versus a certain amount of accuracy for different models on CIFAR-100 dataset.

| Model | CIFAR-100 (40%) | CIFAR-100 (60%) |
|---|---|---|
| **ResNet34** | 1,432,763 | — |
| **MB_ResNet34** | 122,351 | 256,291 |

5.4.2. The Effects of Different Communication Bandwidth Ratios (μ)

To better understand the impact of uniformity using different μ values, we conducted experiments using extreme values in our clustering algorithm. The results are depicted in Figure 10.

By testing these extreme values, we observed that the uniformity performance improved when the computation capability and communication bandwidth were more balanced. Specifically, the extreme values represented by the bottom left and bottom right in Figure 10 demonstrated poorer uniformity performance than the ones on the top. The worst performance case occurred when there was an overindulgence in computation capability. In most federated learning scenarios, the straggler spent significant time in transit, transferring data rather than actively performing computations. This imbalance between computation and communication resulted in a decrease in uniformity performance.

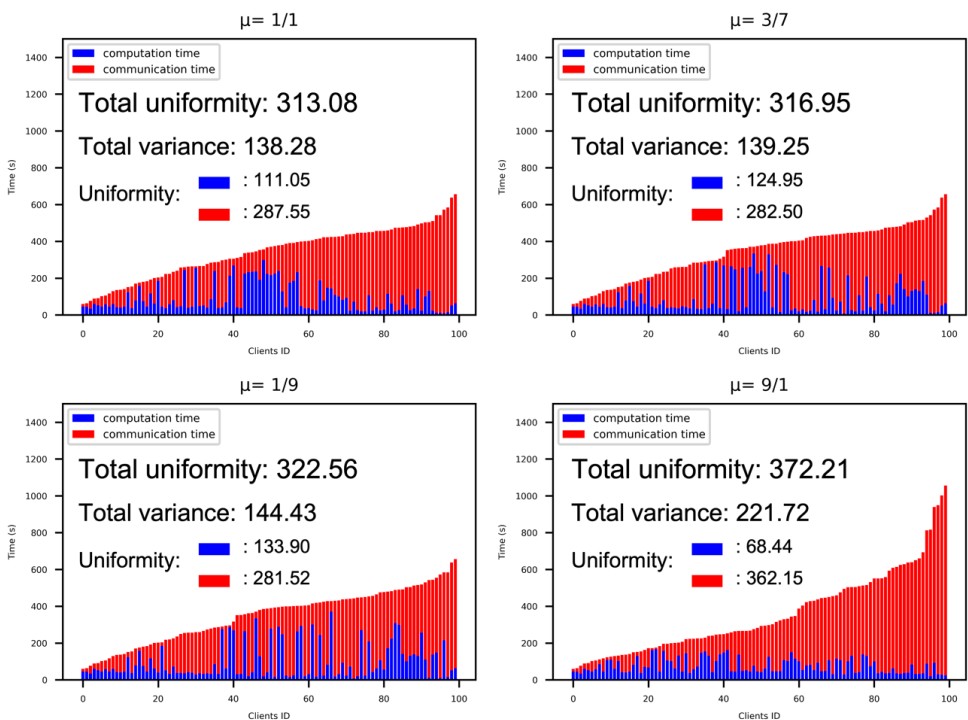

**Figure 10.** The uniformity measures under different values of μ.

## 6. Conclusions

This paper presents an approach integrating a multi-branch network with the Oort client selection algorithm. Our proposed method includes incorporating a Model Distributor module, which efficiently clusters clients and dispatches appropriate models to them. Through extensive experimentation, we have demonstrated the effectiveness of our approach in mitigating the impact of non-iid data, which is not considered in the MFedAvg method. Furthermore, our approach surpasses the performance of the original Oort paper.

Additionally, we introduced the concept of uniformity, which provides a straightforward measure of the training time gap among participants and identifies the presence of stragglers. The concept of uniformity offers valuable insights into the distribution of training time and facilitates the assessment of fairness and efficiency in the federated learning process.

Our results showcase the benefits of integrating a multi-branch network and the Oort client selection algorithm. Furthermore, we have emphasized the significance of considering uniformity in designing and evaluating federated learning frameworks.

In the future, we will conduct further investigations to enhance the overall accuracy of our approach. We plan to explore various techniques, including model architecture modifications, optimization algorithms, and the incorporation of additional data preprocessing methods. Moreover, we will conduct experiments on larger datasets to gain insights into the scalability and generalizability of our approach. We aim to refine and optimize our approach by undertaking these efforts, ultimately achieving higher accuracy and better performance in federated learning. Moreover, as mentioned in Section 3.4, FL plays crucial roles in edge-computing- and IoT-related applications; investigating the applicability of our approach in wireless communication environments is of great interest and will undoubtedly be one of our future research directions.

Finally, as one of the anonymized reviewers pointed out, this work took the bandwidth of the communication channel as one of the criteria for selecting clients; however, the channel characteristics do affect the adequate bandwidth, too. The anonymized reviewer also suggested we consider replacing our current bandwidth measure with the client's achievable data rate. In response to this precious comment, we are searching for practical

datasets (such as those applied to wireless edge FL scenarios) and planning to re-conduct our experiments shortly.

**Author Contributions:** Formal analysis, P.-H.J.; Funding acquisition, J.-L.W.; Investigation, P.-H.J. and J.-L.W.; Methodology, P.-H.J.; Project administration, J.-L.W.; Resources, J.-L.W.; Software, P.-H.J.; Supervision, J.-L.W.; Writing—original draft, P.-H.J.; Writing—review & editing, J.-L.W. All authors have read and agreed to the published version of the manuscript.

**Funding:** This research was partially supported by the Minister of Science and Technology, Taiwan (grant number: MOST 111-2221-E-002-134-MY3), National Taiwan University (grant number: NTU-112L900902), and Taiwan Semiconductor Manufacturing (grant number: TSMC 112H1002-D).

**Data Availability Statement:** No new data were created or analyzed in this study. Data sharing is not applicable to this article.

**Conflicts of Interest:** The authors declare no conflict of interest.

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
