# Peer review of "Enhancing Communication Efficiency and Training Time Uniformity in Federated Learning through Multi-Branch Networks and the Oort Algorithm"

_algorithms, doi:10.3390/a17020052_

Round 1
Reviewer 1 Report
Comments and Suggestions for Authors
This paper combines the multi-branch network and Oort client selection algorithm to balance the communication and computation resources at different nodes in the federated learning scenario. This paper is clearly written and easy to follow. Experiments show that the proposed method is effective. Below are a few comments to improve on the quality of the paper.
1. When reviewing different FL methods to reduce communication burden, only six previous works have been mentioned. In fact, there are many other methods in FL to improve the communication efficiency. One example is [R1], in which over-the-air concept is proposed to make use of the wireless channel property for model aggregation. They are worth to be mentioned in the related work.
2. In general, communication efficiency is important in other learning setting besides FL. A recent example is [R2]. The authors can search for more examples.
3. For the proposed method, it might be effective in handling clients with different communications and computing capabilities. However, it is not clear how the non-i.i.d. data is handled in Section 4. Please add more explanation.
4. For the Oort framework, some users’ data will be used less frequently or even not used at all. Would this affect the learnt model to be very different compared to the ideal case if we gather all data from different users to a central processor for learning the model ? Please comment.
[R1] ``Edge Federated Learning via Unit-modulus Over-the-air Computation," IEEE Trans. on Communications, 2022.
[R2] "Edge learning for large-scale Internet of Things with task-oriented efficient communication," IEEE Transactions on Wireless Communications. 2023.
Author Response
Reply to Reviewer-1 of the paper: Enhancing Communication Efficiency and Training Time Uniformity in Federated Learning through Multi-Branch Networks and the Oort Algorithm.
Comment-1. When reviewing different FL methods to reduce communication burden, only six previous works have been mentioned. In fact, there are many other methods in FL to improve the communication efficiency. One example is [R1], in which over-the-air concept is proposed to make use of the wireless channel property for model aggregation. They are worth to be mentioned in the related work.
Our Reply: Since comments 1 and 2 concern the insufficiency of coverage about related works, we reply to the two comments together. (See our reply to comment 2)
Comment-2. In general, communication efficiency is important in other learning setting besides FL. A recent example is [R2]. The authors can search for more examples.
Our Reply: Thank you for bringing [R1] and [R2] to our attention. We totally agree with the reviewer’s comments about the importance of communication efficiency in edge FL and multiple-task environments; however, we don’t want to distract our discussion focus too much, so we added 10 new references and a new section entitled “3.4 FML in Edge Computing and Wireless Communications” in our revision, as the responses to these valuable comments.
- For the proposed method, it might be effective in handling clients with different communications and computing capabilities. However, it is not clear how the non-i.i.d. data is handled in Section 4. Please add more explanation.
Our Reply: The Oort algorithm utilizes statistical utility criteria to evaluate the data quality of individual clients, encompassing considerations for effectively handling non-i.i.d. data. In each round of training, clients are chosen based on their utility scores, a measure that reflects the relevance and reliability of their data contributions. In response to this data heterogeneity concern, we added one new reference and two new paragraphs in the revision’s Section 3.3 to address the above-mentioned utility measures in more detail.
- For the Oort framework, some users’ data will be used less frequently or even not used at all. Would this affect the learnt model to be very different compared to the ideal case if we gather all data from different users to a central processor for learning the model ? Please comment.
Our Reply: Our answer to this question is: Yes! In the Oort framework, if some users’ data is used less frequently or not used at all, it would affect the learned model compared to the ideal case of gathering all data from different users to a central processor for learning the model. As addressed in our previous reply, the Oort framework considers non-iid data issues and selects user data while discarding biased data. In other words, the Oort framework selects clients based on their data qualities and computing resources for participation in training. Therefore, there will inevitably be a trade-off between training efficiency and accuracy.
[R1] ``Edge Federated Learning via Unit-modulus Over-the-air Computation," IEEE Trans. on Communications, 2022.
[R2] "Edge learning for large-scale Internet of Things with task-oriented efficient communication," IEEE Transactions on Wireless Communications. 2023.
Reviewer 2 Report
Comments and Suggestions for Authors
The paper tackles with federated learning approach that combines a multi-branch network and the Oort client selection algorithm to improve the performance of federated learning systems.
The problem is in details described in the introductory sections. An analysis of the existing literature is provided in section 3. A quick web search reveals interesting works on the paper topic that could be added to this section.
The proposed method in the paper consist in allocate distinct model branches to various clients based on their computational capabilities and communication bandwidth. The paper also introduces the measure of uniformity and analyses the proposed algorithms in terms of uniformity.
Experimental results confirm the efficiency of the proposed algorithms.
You take into account the bandwidth of the communication channel for different clients. But the bandwidth have to be considered with the channel characteristics too. A bad radio channel can significantly reduces the data throughput. The simplest way you can consider is to replacing the bandwidth measure with the client achievable data rate, that includes bandwidth and channel effect.
Author Response
Reply to Reviewer-2 of the paper: Enhancing Communication Efficiency and Training Time Uniformity in Federated Learning through Multi-Branch Networks and the Oort Algorithm.
Comment-1: The paper tackles with federated learning approach that combines a multi-branch network and the Oort client selection algorithm to improve the performance of federated learning systems. The problem is in details described in the introductory sections. An analysis of the existing literature is provided in section 3. A quick web search reveals interesting works on the paper topic that could be added to this section.
Our Reply: Thanks for the valuable comments about extending the coverage of related works. Since the other anonymized reviewer gave a similar suggestion, we included ten new references and one new sub-section in our revision in response to this direction of comments. We focused on the communication efficiency in edge FML over wireless communication environments. Although the addressed topics are still limited, extending our consideration to include wired and wireless use cases is a good start.
Comment-2: You take into account the bandwidth of the communication channel for different clients. But the bandwidth have to be considered with the channel characteristics too. A bad radio channel can significantly reduce the data throughput. The simplest way you can consider is to replacing the bandwidth measure with the client achievable data rate, that includes bandwidth and channel effect.
Our Reply: We totally agree with this valuable comment. The client’s achievable data rate should be a suitable replacement for a simple bandwidth measurement. In response to this precious comment, we are searching for practical datasets (such as those applied to wireless edge FL scenarios) and planning to re-conduct our experiments shortly. However, in preparing such an evaluation, we must investigate appropriate benchmarking works in advance, which takes time. Therefore, we have made this one of our future research topics. We added a paragraph to address this issue at the end of the final section.
Round 2
Reviewer 1 Report
Comments and Suggestions for Authors
This version addressed my previous concerns satisfactorily.